# HyperMiner: Topic Taxonomy Mining with Hyperbolic Embedding

**Yishi Xu, Dongsheng Wang, Bo Chen**[*]**, Ruiying Lu, Zhibin Duan**
National Laboratory of Radar Signal Processing, Xidian University, Xi'an, China
`xuyishi@stu.xidian.edu.cn, bchen@mail.xidian.edu.cn`

**Mingyuan Zhou**
McCombs School of Business, The University of Texas at Austin, USA
`mingyuan.zhou@mccombs.utexas.edu`

## Abstract

Embedded topic models are able to learn interpretable topics even with large and heavy-tailed vocabularies. However, they generally hold the Euclidean embedding space assumption, leading to a basic limitation in capturing hierarchical relations. To this end, we present a novel framework that introduces hyperbolic embeddings to represent words and topics. With the tree-likeness property of hyperbolic space, the underlying semantic hierarchy among words and topics can be better exploited to mine more interpretable topics. Furthermore, due to the superiority of hyperbolic geometry in representing hierarchical data, tree-structure knowledge can also be naturally injected to guide the learning of a topic hierarchy. Therefore, we further develop a regularization term based on the idea of contrastive learning to inject prior structural knowledge efficiently. Experiments on both topic taxonomy discovery and document representation demonstrate that the proposed framework achieves improved performance against existing embedded topic models.

## 1   Introduction

With a long track record of success in a variety of applications [1–6], topic models have emerged as one of the most powerful tools for automatic text analysis. Typically, given a collection of documents, a topic model aims to identify a group of salient topics by capturing common word co-occurrence patterns. Despite their popularity, traditional topic models such as Latent Dirichlet Allocation (LDA) [7] and its variants [8–12] are plagued by complicated posterior inference, presenting a challenge to create deeper and more expressive models of text. Fortunately, recent developments of Variational AutoEncoders (VAEs) and Autoencoding Variational Inference (AVI) [13, 14] have shed light on this problem, resulting in the proposal of a series of Neural Topic Models (NTMs) [15–18]. With better flexibility and scalability, NTMs have gained increasing research interest over the past few years.

Parallel to neural topic modeling, the idea of bringing word embeddings [19, 20] into topic models has also attracted much attention. Considering the large performance degradation over short texts due to limited word co-occurrence information, some early works [21–23] exploit word embeddings as complementary metadata and incorporate them into the generative process of topic models. Recently, more flexible ways [24, 25] of combining word embeddings have been explored thanks to the development of NTMs. For example, Bianchi et al. [26] use word embeddings directly as part of the encoder's input. In particular, a novel one called Embedded Topic Model (ETM) [27] stands out for its performance as well as the elegant way it integrates word embeddings. Specifically, by

---

[*]Corresponding author

36th Conference on Neural Information Processing Systems (NeurIPS 2022).

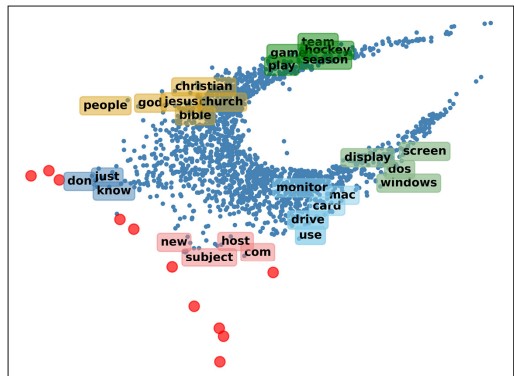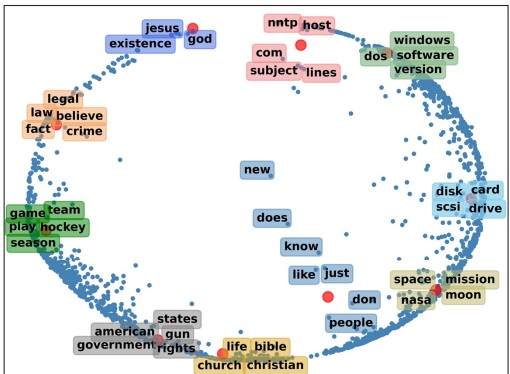

Figure 1: Visualization of 2D Euclidean embedding space (left) and 2D hyperbolic embedding space (right) learned by ETM. Red points denote topic embeddings, blue points represent word embeddings.

representing topics as points in the word embedding space, ETM assigns probabilities to words based on their (inner product) distances from each topic embedding. As a result, semantically related words tend to fall around the same topic, thus facilitating the discovery of more interpretable topics.

Under the inspiration of ETM, Duan et al. [28] have extended a similar idea to hierarchical topic modeling and proposed SawETM. In addition to mapping words and hierarchical topics into a shared embedding space, SawETM has also developed a unique Sawtooth Connection module to capture the dependencies between the topics at different layers, which, on the other side, empowers it to support a deep network structure. While achieving promising results, both ETM and SawETM hold the Euclidean embedding space assumption, leading to a fundamental limitation that their ability to model complex patterns (akin to social networks, knowledge graphs, and taxonomies) is inherently bounded by the dimensionality of the embedding space [29, 30]. As a consequence, the underlying semantic hierarchy among the words and topics can hardly be expressed adequately in a relatively low-dimensional embedding space, as illustrated on the left side of Figure 1.

Apart from the difficulty in capturing the implicit semantic hierarchy, another concomitant problem is the dilemma of incorporating explicit structural knowledge. Assuming we have a prior taxonomy of concepts and wish to use it to guide the learning of hierarchical topics, it is challenging to preserve the structure between concepts in Euclidean space by constraining the word and topic embeddings. To cope with this issue, TopicNet [31] employs the Gaussian-distributed embeddings as a substitute for the vector embeddings to represent words and topics. As such, the prior knowledge of hypernym relations between concepts could be naturally injected via the encapsulation of probability densities. However, maintaining the semantic hierarchy in such an embedding space still suffers from a certain degree of distortion, as it poses a challenge to the optimization of KL divergence between distributions. Furthermore, the introduction of Gaussian-distributed embeddings entails a great demand on memory, limiting its potential scalability to large vocabularies and high-dimensional embedding spaces.

To overcome the above shortcomings brought by Euclidean embedding space, we propose to compute embeddings in hyperbolic space. Distinguished by the tree-likeness properties [32, 33], hyperbolic space has been consistently shown to be superior in modeling hierarchical data compared to Euclidean space [34–37]. By measuring the distance between words and topics in hyperbolic embedding space, the model is encouraged to better capture the underlying semantic hierarchy among words. As shown on the right side of Figure 1, some general words such as "new" and "does" fall around the center, they stay close to all other points because they often co-occur with other words. While more specific words like "moon" and "nasa" fall near the boundary and are only close to the nearby points. Moreover, hyperbolic space also provides a better platform to inject prior structural knowledge, since hierarchical relations can be effectively preserved by imposing constraints on the distance between word and topic embeddings. In a nutshell, the main contributions of this paper are as follows:

- We propose to compute the distance between topics and words in hyperbolic embedding space on top of existing embedded topic models, which is beneficial to both the mining of implicit semantic hierarchy and the incorporation of explicit structural knowledge.

- We design a node-level graph representation learning scheme that can inject prior structural knowledge to effectively guide the learning of a meaningful topic taxonomy.

- Extensive experiments on topic quality and document representation demonstrate that the proposed approach achieves competitive performance against baseline methods.

## 2 Background

### 2.1 Embedded topic model

ETM [27] is a neural topic model that builds on two main techniques: LDA [7] and word embeddings [19, 20]. To marry the probabilistic topic modeling of LDA with the contextual information brought by word embeddings, ETM maintains vector representations of both words and topics and uses them to derive the per-topic distribution over the vocabulary. Specifically, consider a corpus with $V$ distinct terms comprising the vocabulary, we denote the word embedding matrix as $\boldsymbol{\rho} \in \mathbb{R}^{D \times V}$, where $D$ is the dimensionality of the embedding space. For each topic, there is also an embedding representation $\boldsymbol{\alpha}_k \in \mathbb{R}^D$, then ETM defines the per-topic distribution $\boldsymbol{\beta}_k \in \mathbb{R}^V$ over the vocabulary as

$$\boldsymbol{\beta}_k = \text{Softmax}\left(\boldsymbol{\rho}^\top \boldsymbol{\alpha}_k\right) \tag{1}$$

With the above definition, ETM specifies a generative process analogous to LDA. Let $\boldsymbol{w}_{jn} \in \{1, ..., V\}$ denote the $n^{th}$ word in the $j^{th}$ document, the generative process is as follows.

1. Draw topic proportions $\boldsymbol{\theta}_j \sim \mathcal{LN}(\mathbf{0}, \boldsymbol{I})$.
2. For each word $n$ in the document:
   (a) Draw topic assignment $\boldsymbol{z}_{jn} \sim \text{Cat}(\boldsymbol{\theta}_j)$.
   (b) Draw word $\boldsymbol{w}_{jn} \sim \text{Cat}(\boldsymbol{\beta}_{\boldsymbol{z}_{jn}})$.

Where $\mathcal{LN}(\cdot)$ in step 1 denotes the logistic-normal distribution [38], which transforms a standard Gaussian random variable to the simplex. By taking the inner product of the word embedding matrix $\boldsymbol{\rho}$ and the topic embedding $\boldsymbol{\alpha}_k$ to derive $\boldsymbol{\beta}_k$, the intuition behind ETM is that semantically related words will be assigned to similar topics. With this property, ETM has been demonstrated to improve the quality of the learned topics, especially in the presence of large vocabularies. Like most NTMs, ETM is fitted via an efficient amortized variational inference algorithm.

### 2.2 Hyperbolic geometry

In this part, we briefly review some key concepts on hyperbolic geometry. A comprehensive and in-depth description can be found in Lee [39] and Nickel and Kiela [40]. Under the mathematical framework of Riemannian geometry, hyperbolic geometry specializes in the case of constant negative curvatures. Intuitively, the hyperbolic space can be understood as a continuous version of trees: the volume of a ball expands exponentially with its radius, just as how the number of nodes in a binary tree grows exponentially with its depth. Mathematically, there exist multiple equivalent models for hyperbolic space with different definitions and metrics. Here, we consider two representative ones in light of optimization simplicity and stability: Poincaré ball model [29] and the Lorentz model [40].

**Poincaré ball model**   The Poincaré ball model of an $n$-dimensional hyperbolic space with curvature $C$ $(C < 0)$ is defined by the Riemannian manifold $\mathcal{P}^n = (\mathcal{B}^n, g_p)$, where $\mathcal{B}^n = \{\boldsymbol{x} \in \mathbb{R}^n : \|\boldsymbol{x}\| < 1/\sqrt{|C|}\}$ is the open $n$-dimensional ball with radius $1/\sqrt{|C|}$ and $g_p$ is the metric tensor that can be converted from the Euclidean metric tensor $g_e = I$ as

$$g_p(\boldsymbol{x}) = \left(\frac{2}{1 + C\|\boldsymbol{x}\|^2}\right)^2 g_e \tag{2}$$

**Lorentz model**   The Lorentz model (also named Hyperboloid model) of an $n$-dimensional hyperbolic space with curvature $C$ $(C < 0)$ is defined by the Riemannian manifold $\mathcal{L}^n = (\mathcal{H}^n, g_l)$, where $\mathcal{H}^n = \{\boldsymbol{x} \in \mathbb{R}^{n+1} : \langle \boldsymbol{x}, \boldsymbol{x} \rangle_{\mathcal{L}} = 1/C\}$ and $g_l = \text{diag}\left([-1, \mathbf{1}_n^\top]\right)$. $\langle \cdot, \cdot \rangle_{\mathcal{L}}$ denote the *Lorentzian inner product*. Let $\boldsymbol{x}, \boldsymbol{y} \in \mathbb{R}^{n+1}$, the Lorentz inner product induced by $g_l$ is calculated as

$$\langle \boldsymbol{x}, \boldsymbol{y} \rangle_{\mathcal{L}} = \boldsymbol{x}^\top g_l \boldsymbol{y} = -x_0 y_0 + \sum_{i=1}^{n} x_i y_i \tag{3}$$

An intuitive illustration of the equivalence between the Poincaré ball model and the Lorentz model and some other related operations in hyperbolic space will be introduced in Appendix C.

# 3   Topic Taxonomy Mining with Hyperbolic Embedding

In this section, we elaborate on how the introduced hyperbolic embeddings facilitate the mining of implicit semantic hierarchy and the incorporation of explicit tree-structure knowledge, both of which encourage the model to find more interpretable topics[2].

## 3.1   Hierarchical topic modeling in hyperbolic space

Theoretically, the idea of representing words and topics in hyperbolic space is orthogonal to a wide range of topic models employing the word embeddings technique. To provide the foundation for the subsequent injection of structural knowledge, we here apply our method to a hierarchical embedded topic model SawETM [28]. SawETM utilizes the adapted Poisson gamma belief network (PGBN) [41] as its generative module (decoder) and decomposes the topic matrices into the inner product of topic embeddings at adjacent layers. The novelty of our method lies in that the hierarchical relations can be better reflected by the distances between embeddings in hyperbolic space. Mathematically, the generative model with $L$ latent layers is formulated as

$$
\begin{aligned}
&\boldsymbol{\theta}_j^{(L)} \sim \mathrm{Gam}\left(\boldsymbol{\gamma}, e_j^{(L+1)}\right), \boldsymbol{\theta}_j^{(l)} \sim \mathrm{Gam}\left(\boldsymbol{\Phi}^{(l+1)}\boldsymbol{\theta}_j^{(l+1)}, e_j^{(l+1)}\right), \ldots, \\
&\boldsymbol{\theta}_j^{(1)} \sim \mathrm{Gam}\left(\boldsymbol{\Phi}^{(2)}\boldsymbol{\theta}_j^{(2)}, e_j^{(2)}\right), \boldsymbol{x}_j \sim \mathrm{Pois}\left(\boldsymbol{\Phi}^{(1)}\boldsymbol{\theta}_j^{(1)}\right), \\
&\boldsymbol{\Phi}^{(l)} = \mathrm{Softmax}\left(\mathcal{S}\left(\boldsymbol{\alpha}^{(l-1)}, \boldsymbol{\alpha}^{(l)}\right)\right)
\end{aligned}
\tag{4}
$$

The above formula clearly describes how the multi-layer document representation is generated via a top-down process. Specifically, the latent representation of the top layer $\boldsymbol{\theta}_j^{(L)}$ is sampled from a fixed gamma prior distribution, then at each intermediate layer $l$ the latent units $\boldsymbol{\theta}_j^{(l)} \in \mathbb{R}^{K_l}$ are factorized into the product of the factor loading matrix $\boldsymbol{\Phi}^{(l+1)} \in \mathbb{R}^{K_l \times K_{l+1}}$ and latent units $\boldsymbol{\theta}_j^{(l+1)} \in \mathbb{R}^{K_{l+1}}$ of the above layer. Until the bottom layer, the observation of word count vector $\boldsymbol{x}_j \in \mathbb{Z}^V$ is modeled as the Poisson distribution. Note that the subscript $j$ denotes the document index and some other variables $\boldsymbol{\gamma}, e_j^{(L+1)}, \ldots, e_j^{(2)}$ are hyperparameters. Especially, the factor loading matrix $\boldsymbol{\Phi}^{(l)}$ of layer $l$ is derived based on the distance between the topic embeddings at two adjacent layers, i.e., $\boldsymbol{\alpha}^{(l-1)} \in \mathbb{R}^{K_{l-1} \times D}$ and $\boldsymbol{\alpha}^{(l)} \in \mathbb{R}^{K_l \times D}$. Note that $\boldsymbol{\alpha}^{(0)} \in \mathbb{R}^{V \times D}$ represents the word embeddings. Since all embeddings are projected into the hyperbolic space to fully explore the underlying semantic hierarchy among the words and topics, we design our similarity score function as

$$
\begin{aligned}
&\mathcal{S}(\boldsymbol{x}, \boldsymbol{y}) = -d_{\mathcal{P}}(\boldsymbol{x}, \boldsymbol{y}) = \frac{-1}{\sqrt{|C|}}\mathrm{arcosh}\left(1 - \frac{2C\|\boldsymbol{x}-\boldsymbol{y}\|^2}{\left(1+C\|\boldsymbol{x}\|^2\right)\left(1+C\|\boldsymbol{y}\|^2\right)}\right) \\
&\mathcal{S}(\boldsymbol{x}, \boldsymbol{y}) = -d_{\mathcal{L}}(\boldsymbol{x}, \boldsymbol{y}) = \frac{-1}{\sqrt{|C|}}\mathrm{arcosh}\left(C\langle\boldsymbol{x}, \boldsymbol{y}\rangle_{\mathcal{L}}\right)
\end{aligned}
\tag{5}
$$

Where $d_{\mathcal{P}}(\cdot, \cdot)$ and $d_{\mathcal{L}}(\cdot, \cdot)$ are the distance functions of the Poincaré ball model and the Lorentz model, respectively. As the two models of hyperbolic space are mathematically equivalent, we take the Poincaré ball as an example for analysis. Eq. (5) shows that the distance changes smoothly with respect to the norm of $\boldsymbol{x}$ and $\boldsymbol{y}$. This locality plays a crucial role in learning continuous embeddings of hierarchies. For instance, the origin of $\mathcal{B}^n$ has a zero norm, it would have relatively small distance to all other points, which exactly corresponds to the root node of a tree. On the other hand, those points close to the boundary of the ball have a norm close to one, so the distance between them grows quickly, which properly reflects the relationships between the leaf nodes of a tree.

## 3.2   Knowledge guided topic taxonomy discovery

Hierarchical structures are ubiquitous in knowledge representation and reasoning [40]. Particularly, mining a set of meaningful topics organized into a hierarchy from massive text corpora is intuitively appealing, as it allows users to easily access the information of their interest. However, most existing hierarchical topic models struggle to realize this goal without any supervision, and some appropriate guidance with prior structural knowledge proves to be helpful for mitigating this issue [31, 42].

---

[2]Our code is available at `https://github.com/NoviceStone/HyperMiner`

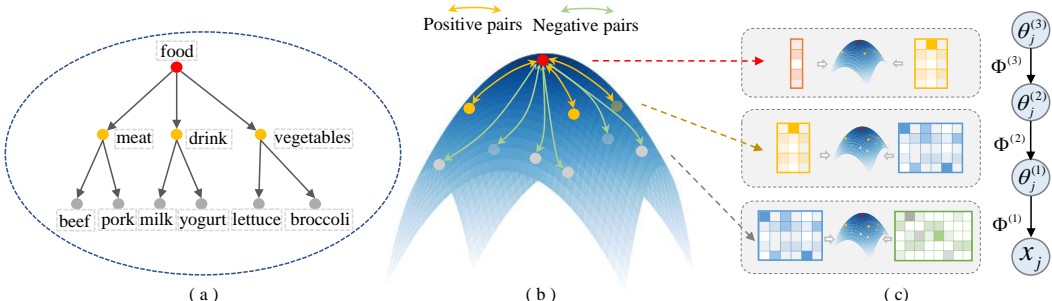

Figure 2: Overview of knowledge guided topic taxonomy discovery. (a) The prior concept taxonomy constructed from vocabulary; (b) Illustration of the strategy for picking positive pairs and negative pairs in the hyperbolic embedding space; (c) The hierarchical generative model whose factor loading matrices are derived based on the hyperbolic distances between topic embeddings at two adjacent layers.

**Form of prior knowledge**    We assume the prior knowledge takes the form of a concept taxonomy, which is compatible with the deep structure of the proposed hierarchical topic model in Section 3.1. In detail, the taxonomy exhibits a top-down reification process and concepts between two adjacent layers with connections follow the hypernym relation, as shown in Figure 2(a). Meanwhile, to keep the taxonomy consistent with the corresponding dataset, we construct it by traversing each word in the vocabulary to find its ancestors along the hypernym paths provided by WordNet [43].

**Hyperbolic contrastive loss**    Although the paradigm of contrastive learning has been successfully applied to graph representation learning in Euclidean space [44–47], those developed contrastive algorithms are not directly applicable in our case. On the one hand, we focus more on learning node representations while not destroying their prior hierarchical structures. On the other hand, hyperbolic space possesses distinctive properties (*e.g.*, hierarchical awareness and spacious room) compared to its Euclidean counterpart. Consequently, to accommodate the two differences, we design a node-level hyperbolic contrastive loss such that the prior knowledge can be effectively injected as an inductive bias to influence the learning of topic taxonomy.

Specifically, we set the number of topics at each layer of the hierarchical topic model to be the same as the number of concepts at each layer of the taxonomy, then each topic is assigned a corresponding concept as its semantic prior. Since concepts are connected in the taxonomy, such relations can be transfered accordingly between topics, which provide the basis for picking positive and negative pairs among the topic and word embeddings. Let $\mathcal{T} = \{\boldsymbol{\alpha}^{(l)}\}_{l=0}^{L}$ denote the set of all embeddings, then each embedding $\boldsymbol{\alpha}_i^{(l)} \in \mathbb{R}^D$ is associated with two groups of embeddings as the positive samples and the negative samples, respectively. Then the average hyperbolic contrastive loss is defined as (we omit the superscript $l$ for simplicity of notation)

$$\mathcal{L}_{\text{Contra}} = \mathbb{E}_{\boldsymbol{\alpha}_i \in \mathcal{T}} \left[ -\log \frac{\exp\left(\mathcal{S}(\boldsymbol{\alpha}_i, \boldsymbol{\alpha}_i^+)/\tau\right)}{\exp\left(\mathcal{S}(\boldsymbol{\alpha}_i, \boldsymbol{\alpha}_i^+)/\tau\right) + \sum_{\boldsymbol{\alpha}_i^- \in \mathcal{Q}(\boldsymbol{\alpha}_i)} \exp\left(\mathcal{S}(\boldsymbol{\alpha}_i, \boldsymbol{\alpha}_i^-)/\tau\right)} \right] \tag{6}$$

where $\mathcal{S}(\cdot, \cdot)$ is the similarity score function defined in Eq. (5) and $\tau$ is the temperature parameter. Note that $\boldsymbol{\alpha}_i^+$ is a positive sample drawn from $\mathcal{P}(\boldsymbol{\alpha}_i)$ and $\mathcal{Q}(\boldsymbol{\alpha}_i)$ is the set of negative samples.

**Sampling strategy**    Inspired by the homophily property (*i.e.*, similar actors tend to associate with each other) in many graph networks [29], we take one-hop neighbors of each anchor, *i.e.*, its parent node and its child nodes as positive samples to maintain the hierarchical semantic information. For the negative samples, we select $m$ embeddings from the non-first-order neighbors that have the highest similarity scores with the anchor embedding.

### 3.3   Training objective

As most existing NTMs can be viewed as the extensions of the framework of VAEs [13, 14], they generally develop a similar training objective to VAEs, which is to maximize the Evidence Lower BOund (ELBO). For our generative model, the ELBO of each document can be derived as

$$\mathcal{L}_{\text{ELBO}} = -\sum_{l=1}^{L} D_{KL}\left[q(\boldsymbol{\theta}_j^{(l)}|-)\|p(\boldsymbol{\theta}_j^{(l)}|\boldsymbol{\Phi}^{(l+1)}, \boldsymbol{\theta}_j^{(l+1)})\right] + \mathbb{E}_{q(\boldsymbol{\theta}_j^{(1)}|-)}\left[\ln p(\boldsymbol{x}_j|\boldsymbol{\Phi}^{(1)}, \boldsymbol{\theta}_j^{(1)})\right] \tag{7}$$

---
**Algorithm 1** Knowledge-Guided Topic Taxonomy Mining
---
Input: mini-batch size $B$, number of layers $T$, adjacent matrix $A$ built from concept taxonomy.
Initialize the variational network parameters $\boldsymbol{\Omega}$ and the word and topic embeddings $\{\boldsymbol{\alpha}^{(l)}\}_{l=0}^{L}$;
**while** not converged **do**

    1. Randomly draw a batch of samples $\{\boldsymbol{x}_j\}_{j=1}^{B}$;

    2. Infer variational posteriors for the latent variables of different layers $\{\boldsymbol{\theta}_j^{(l)}\}_{j=1,l=1}^{B,L}$;

    3. Derive factor loading matrices $\{\boldsymbol{\Phi}^{(l)}\}_{l=1}^{L}$ using $\{\boldsymbol{\alpha}^{(l)}\}_{l=0}^{L}$ based on Eq. (4);

    4. Compute the ELBO on the joint marginal likelihood of $\{\boldsymbol{x}_j\}_{j=1}^{B}$ based on Eq. (7);

    5. Compute the hyperbolic contrastive loss using $\{\boldsymbol{\alpha}^{(l)}\}_{l=0}^{L}$ and $A$ according to Eq. (6);

    6. Update $\boldsymbol{\Omega}$ and $\{\boldsymbol{\alpha}^{(l)}\}_{l=0}^{L}$ using gradients $\nabla_{\boldsymbol{\Omega},\boldsymbol{\alpha}^{(l)}}\mathcal{L}\left(\boldsymbol{\Omega},\{\boldsymbol{\alpha}^{(l)}\}_{l=0}^{L};\{\boldsymbol{x}_j\}_{j=1}^{B}\right)$;

**end while**
---

where the first term is the Kullback–Leibler divergence that constrains the approximate posterior $q(\boldsymbol{\theta}_j^{(l)}|-)$ to be close to the prior $p(\boldsymbol{\theta}_j^{(l)})$, and the second term denotes the expected log-likelihood or reconstruction error. Considering that our generative model employs the gamma-distributed latent variables, it brings the difficulty of reparameterizing a gamma-distributed random variable when we design a sampling-based inference network. Therefore, we instead utilize a Weibull distribution to approximate the conditional posterior inspired by Zhang et al. [17], as the analytic KL expression and efficient reparameterization make it easy to estimate the gradient of ELBO with respect to network parameters. The implementation details of our variational encoder is described in Appendix B.

Furthermore, to inject the prior knowledge to guide the learning of a topic taxonomy, we train the ELBO jointly with a regularization term specified by the proposed contrastive loss in Section 3.2

$$\mathcal{L} = \mathcal{L}_{\text{ELBO}} + \lambda\mathcal{L}_{\text{Contra}} \tag{8}$$

where $\lambda$ is the hyper-parameter used to control the impact of the regularization term, whose detailed effect is investigated in Appendix D. We summarize our complete learning procedure in Algorithm 1.

Table 1: Statistics of the datasets

|        | Number of docs | Vocabulary size | Total number of words | Categories |
|--------|----------------|-----------------|-----------------------|------------|
| 20NG   | 18,846         | 8,581           | 1,446,490             | 20         |
| TMN    | 32,597         | 13,368          | 592,973               | 7          |
| WIKI   | 28,472         | 20,000          | 3,719,096             | N/A        |
| RCV2   | 804,414        | 7,282           | 60,209,009            | N/A        |

## 4 Experiments

### 4.1 Experimental setup

**Datasets**  We conduct our experiments on four benchmark datasets with various sizes and document lengths, including *20Newsgroups* (**20NG**) [48], *Tag My News* (**TMN**) [49], *WikiText-103* (**WIKI**) [50], and *Reuters Corpus Volume II* (**RCV2**) [51]. The statistics of these datasets are presented in Table 1. In particular, TMN is a short text corpus with an average of about 20 words per document; 20NG and TMN are the two corpora that are associated with document labels.

**Baseline methods**  As baselines, we choose several exemplary ones from the state-of-the-art topic models, including: 1) **LDA** [7], one of the most widely used topic models; 2) **ProdLDA** [16], an NTM which replaces the mixture model in LDA with a product of experts; 3) **ETM** [27], an NTM that marries conventional topic models with word embeddings; 4) **WHAI** [17], a hierarchical NTM which develops a deep Weibull variational encoder based on PGBN [41]; 5) **SawETM** [28], which proposes a Sawtooth Connection module to build the dependencies between topics at different layers; 6) **TopicNet** [31], a knowledge-based hierarchical NTM that guides topic discovery through prior semantic graph. All baselines are implemented meticulously according to their official code.

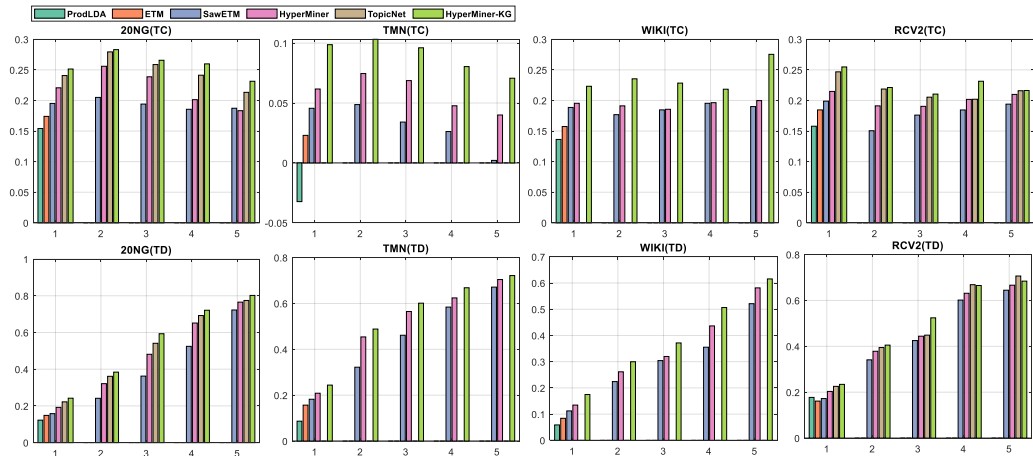

Figure 3: The performance comparison of different models on the topic quality. The top row shows the topic coherence score, $i.e.$, NPMI, and the bottom row displays the topic diversity score. The horizontal axis represents the index of the layers and we set up 5 layers for all the hierarchical topic models. The result of TopicNet on TMN and WIKI is missing because of memory overflow for large vocabulary size.

**Evaluation metrics**  We aim to evaluate our model's performance in terms of both topic quality and document representation. For topic quality, we adopt topic coherence (**TC**) and topic diversity (**TD**) as performance metrics. Given a reference corpus, TC measures the interpretability of each topic by computing the semantic coherence of the most significant words [52]. Precisely, we apply the widely used Normalized Pointwise Mutual Information (NPMI) [53] and compute it over the top 10 words of each topic, with the original document collections of each dataset serving as the reference corpus. Note that the value of NPMI ranges from $-1$ to 1, and higher values indicate better interpretability. TD, as the name suggests, measures how diverse the discovered topics are. Following Dieng et al. [27], we define TD to be the percentage of unique words in the top 25 words of all learned topics. TD close to 0 means redundant topics and TD close to 1 implies more varied topics.

On the other hand, since per-document topic proportions can be viewed as unsupervised document representations, we intend to evaluate the quality of such representations by performing document clustering tasks. We report the purity and Normalized Mutual Information (NMI) [54] on two datasets providing the document labels, $i.e.$, 20NG and TMN. Concretely, with the default training/test split of each dataset, we first train a topic model on the training set, and then the trained model is used to extract features $\boldsymbol{\theta}$ of all test documents. Subsequently, we apply the KMeans algorithm on $\boldsymbol{\theta}$ and calculate the purity and NMI of the KMeans clusters (denoted by **km-Purity** and **km-NMI**). Note that both of the two metrics range from 0 to 1, and higher scores indicate better performance. For the hierarchical topic models, we take latent units $\boldsymbol{\theta}^{(1)}$ of the first layer as the document feature.

## 4.2 Experimental results

**Topic quality**  Considering that not all discovered topics are interpretable [55], we select the top 50% topics with the highest NPMI values and report the average score over those selected topics to evaluate the topic quality comprehensively. Figure 3 exhibits the performance comparison results of different models. Note that HyperMiner is the variant of SawETM that replaces the inner product between Euclidean embeddings with the distance between hyperbolic embeddings, corresponding to the model proposed in Section 3.1. While HyperMiner-KG is an advanced version of HyperMiner that guides the learning of topic taxonomy by external structural knowledge, as introduced in Section 3.2. From what is shown in Figure 3, HyperMiner achieves consistent performance gains on all datasets compared to SawETM, in regard of both TC and TD, which demonstrates the superiority of hyperbolic geometry in uncovering the latent hierarchies among topics and words. In addition, as knowledge-guided topic models, both TopicNet and HyperMiner-KG get better performance than those without any supervision, indicating the positive role of prior knowledge in helping to mine more interpretable and diverse topics. However, HyperMiner-KG still performs slightly better than TopicNet while consuming less memory. We attribute this result to our naturally-designed framework of injecting the tree-structure knowledge in a contrastive manner.

Table 2: km-Purity and km-NMI for document clustering. The best and second best scores of each dataset are highlighted in boldface and with an underline, respectively. The embedding dimension for embedded topic models is set as 50.

| Method | 20NG | | TMN | |
|---|---|---|---|---|
| | km-Purity | km-NMI | km-Purity | km-NMI |
| LDA [7] | $38.43 \pm 0.52$ | $35.98 \pm 0.39$ | $48.17 \pm 0.69$ | $30.96 \pm 0.78$ |
| ProdLDA [16] | $39.21 \pm 0.63$ | $36.52 \pm 0.51$ | $55.28 \pm 0.67$ | $35.57 \pm 0.72$ |
| ETM [27] | $42.68 \pm 0.71$ | $37.72 \pm 0.64$ | $59.35 \pm 0.74$ | $38.75 \pm 0.86$ |
| WHAI [17] | $40.89 \pm 0.35$ | $38.90 \pm 0.27$ | $58.06 \pm 0.45$ | $37.34 \pm 0.48$ |
| SawETM [28] | $43.36 \pm 0.48$ | $41.59 \pm 0.62$ | $61.13 \pm 0.56$ | $40.78 \pm 0.63$ |
| TopicNet [31] | $42.94 \pm 0.41$ | $40.76 \pm 0.53$ | $60.52 \pm 0.50$ | $40.09 \pm 0.54$ |
| HyperETM | $43.63 \pm 0.51$ | $39.06 \pm 0.64$ | $61.22 \pm 0.62$ | $40.52 \pm 0.71$ |
| HyperMiner | $\underline{44.37} \pm 0.38$ | $\underline{42.83} \pm 0.45$ | $\underline{62.96} \pm 0.48$ | $\underline{41.93} \pm 0.52$ |
| HyperMiner-KG | $\mathbf{45.16} \pm 0.35$ | $\mathbf{43.65} \pm 0.39$ | $\mathbf{63.84} \pm 0.43$ | $\mathbf{42.81} \pm 0.47$ |

Table 3: Accuracy for document classification on 20NG, with different embedding dimensions for embedded topic models.

| | $D = 2$ | $D = 5$ | $D = 10$ | $D = 20$ | $D = 50$ |
|---|---|---|---|---|---|
| ETM [27] | $19.87 \pm 0.81$ | $33.64 \pm 0.69$ | $39.06 \pm 0.54$ | $42.13 \pm 0.47$ | $43.85 \pm 0.42$ |
| HyperETM | $\mathbf{24.33} \pm 0.76$ | $\mathbf{36.57} \pm 0.65$ | $\mathbf{40.92} \pm 0.56$ | $\mathbf{43.04} \pm 0.43$ | $\mathbf{44.38} \pm 0.40$ |
| SawETM [28] | $16.74 \pm 0.78$ | $27.05 \pm 0.66$ | $31.68 \pm 0.51$ | $34.06 \pm 0.42$ | $35.42 \pm 0.37$ |
| HyperMiner | $\mathbf{20.16} \pm 0.80$ | $\mathbf{29.73} \pm 0.63$ | $\mathbf{33.04} \pm 0.49$ | $\mathbf{34.98} \pm 0.41$ | $\mathbf{36.01} \pm 0.36$ |
| TopicNet [31] | $20.29 \pm 0.58$ | $31.26 \pm 0.51$ | $34.57 \pm 0.45$ | $36.84 \pm 0.39$ | $38.02 \pm 0.36$ |
| HyperMiner-KG | $\mathbf{22.83} \pm 0.55$ | $\mathbf{33.15} \pm 0.50$ | $\mathbf{36.28} \pm 0.43$ | $\mathbf{38.11} \pm 0.40$ | $\mathbf{39.46} \pm 0.34$ |

**Document representation**   Table 2 shows the clustering performance of different models. We run all the models in comparison five times with different random seeds and report the mean and standard deviation. From the results presented above, we have the following remarks: **i)** For all the evaluation metrics, our proposed improved variants perform consistently better than their prototypical models (refer to HyperETM versus ETM, and HyperMiner versus SawETM), which demonstrates that the introduced hyperbolic embeddings are beneficial to both the discovery of high-quality topics and the learning of good document representations. **ii)** As a knowledge-guided topic model, HyperMiner-KG achieves a significant improvement over the base model SawETM, while TopicNet suffers a slight performance degradation compared to SawETM, which also serves as its base model in the original paper. This observation shows that with the hyperbolic contrastive loss, our model not only injects the knowledge successfully into the learning of hierarchical topics, but also achieves a better balance among the comprehensive metrics of topic modeling. **iii)** The superior performance of our model on TMN also suggests its potential for short text topic modeling.

To further investigate the effectiveness of our method under different dimensional settings, we proceed to compare the extrinsic predictive performance of document representations through classification tasks. Consistent with the practice in clustering tasks, we first collect the features of training set $\boldsymbol{\theta}_{tr}$ and test set $\boldsymbol{\theta}_{te}$ separately, which are inferred by a well-trained topic model. Then we train an SVM classifier using $\boldsymbol{\theta}_{tr}$ and their corresponding labels. Finally, we use the trained classifier to predict the labels of $\boldsymbol{\theta}_{te}$ and compute the accuracy. Table 3 illustrates the classification results of different embedded topic models. From the table we can see that the improved variants with our method surpass their base counterparts in various dimensionality settings. Especially, the performance gap between them has been further widened in the low-dimensional embedding space, confirming the natural advantage of hyperbolic distance metric in learning useful document representations.

**Visualization of embedding space and topics**   As our proposed HyperMiner-KG imposes a prior hierarchical constraint (*i.e.*, concept taxonomy) on the embedding space, the topic embeddings and

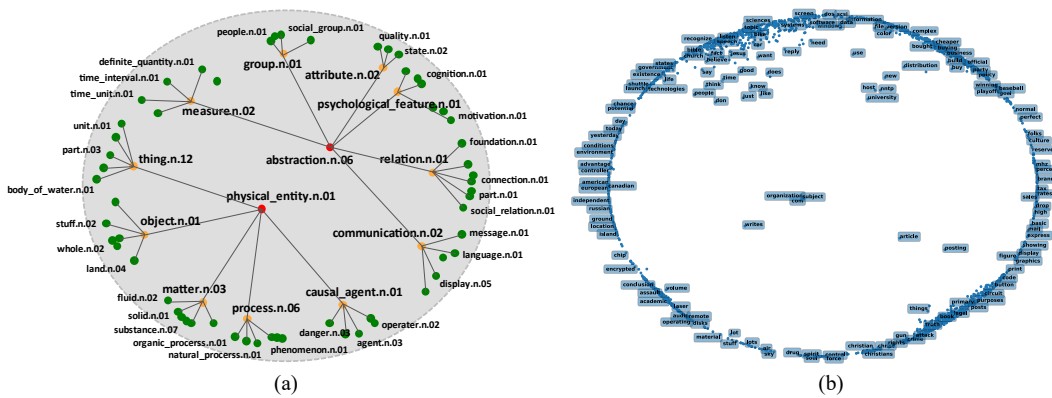

Figure 4: Visualization of 2D hyperbolic embedding space learned by HyperMiner-KG. (a) concept hierarchy: topic embeddings and their corresponding prior semantic concepts, where different coloured points represent topic embeddings at different layers. (b) lexical hierarchy: word embeddings and their corresponding meanings.

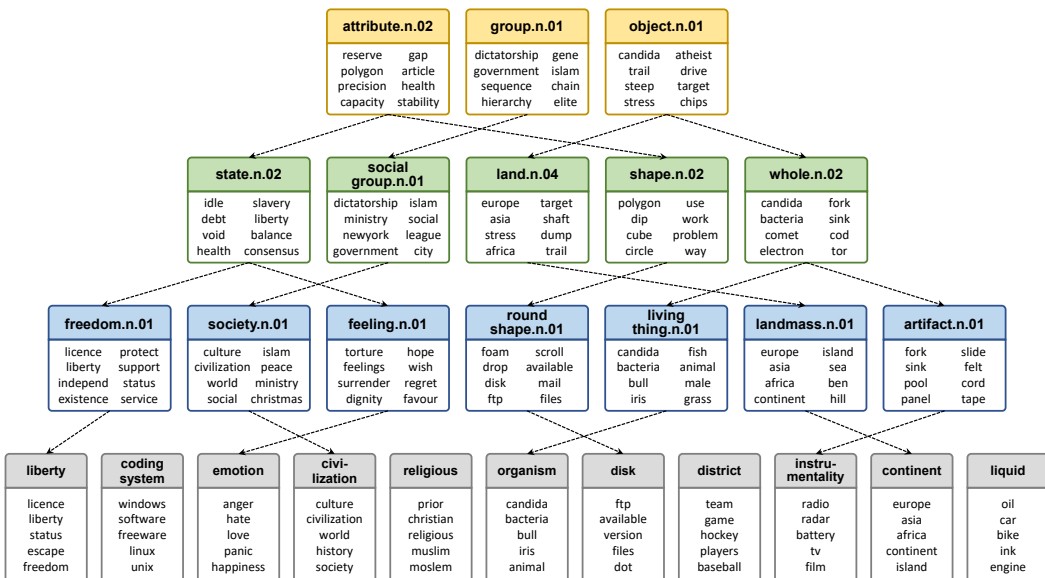

Figure 5: Illustration of the topic taxonomy learned by a 5-layer HyperMiner-KG on *20Newsgroups*, we show the contents of some example topics, which are learned under the guidance of prior semantic concepts. Note that the example topics are selected from the bottom 4 layers, with different colored boxes to distinguish them.

word embeddings are learned to maintain this structure as much as possible. Therefore, to verify the effectiveness of our proposed regularization term for injecting structural knowledge, we visualize the two-dimensional hyperbolic embedding space learned by HyperMiner-KG, as displayed in Figure 4. Figure 4(a) exhibits the learned topic embeddings and the concept hierarchy used to guide them, in which we can see that the distribution of topic embeddings well preserves the semantic structure of prior knowledge. Specifically, for topics guided by higher-level and more general concepts (*e.g.*, *physical_entity.n.01*), their embeddings tend to locate in the center of the disc. While for those led by slightly more certain concepts (*e.g.*, *substance.n.07*), their embeddings prefer to scatter around the boundary. Figure 4(b) presents the distribution of learned word embeddings, which also reflects the underlying lexical hierarchy of the corpus. The words that co-occur more frequently with different terms (*e.g.*, *organization*, *subject*) tend to fall around the center of the disc, so as to maintain a small distance from arbitrary words. In contrast, those words with precise meanings fall near the edge area with spacious room and keep a small distance only from words with similar semantics.

Furthermore, to qualitatively demonstrate the crucial role of prior structural knowledge in helping to discover more interpretable topics, we show the contents (*i.e.*, top words) of some learned topics by HyperMiner-KG, as illustrated in Figure 5. From it we can observe that in a majority of cases the

prior concepts can successfully guide the topics to learn semantically consistent contents (*e.g.*, the topics guided by *coding_system* and *instrumentality*). Moreover, the contents of topics at different layers are also semantically related due to the concepts guiding them. For instance, the content of the topic guided by *whole.n.02* covers the contents of topics led by *living_thing.n.01* and *artifact.n.01*, respectively. Another interesting phenomenon is that the topic led by *round_shape.n.01* involves not only words related to shapes, but also some other words such as *files* and *ftp*. The reason could be one of its child concepts *disk* often co-occurs with those words in the given corpus, suggesting the topic learning is co-influenced by both data likelihood and knowledge regularization.

## 5 Related Work

Historically, many attempts have been made to develop hierarchical topic models. Expanding on their flat counterparts, hierarchical topic models aim to generate an intuitive topic hierarchy by capturing the correlations among topics. Due to the inflexibility of requiring accurate posterior inference, early works [8, 9, 12, 41] primarily focused on learning potential topic hierarchies purely from data, with some additional probabilistic assumptions being imposed. Also, there is a small body of work that tries to integrate domain knowledge into the process of discovering topic hierarchies. For example, anchored CorEx [56] takes user-provided seed words and learns informative topics by maximizing total correlation while preserving anchor words related information. More recently, JoSH [57] adopts a more effective strategy that takes a category hierarchy as the guidance and models category-word semantic correlation via joint spherical text and tree embedding. Different from anchored CorEx and JoSH, which deviate from the conventional topic modeling framework, our approach still follows the regular probabilistic generative process. In addition, we use a concept taxonomy to guide the topic learning so that more fine-grained topics can be mined. In this regard, our proposed HyperMiner-KG is much more related to TopicNet [31], yet it is more efficient with a smaller storage footprint.

## 6 Conclusion

This paper presents a novel framework that introduces hyperbolic embeddings to represent words and topics on top of existing embedded topic models. By using the hyperbolic distance to measure the semantic similarity between words and topics, the model can better explore the underlying semantic hierarchy to find more interpretable topics. Besides, a hyperbolic contrastive loss has been further proposed, which effectively injects prior structural knowledge into hierarchical topic models to guide learning a meaningful topic taxonomy. Our method shows appealing properties that can overcome several shortcomings of existing embedded topic models. Extensive experiments have been carried out, demonstrating that our method achieves consistent performance improvements in discovering high-quality topics and deriving useful document representations.

### Acknowledgments and Disclosure of Funding

This work was supported in part by the National Natural Science Foundation of China under Grant U21B2006; in part by Shaanxi Youth Innovation Team Project; in part by the 111 Project under Grant B18039; in part by the Fundamental Research Funds for the Central Universities QTZX22160.

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
