# Supplementary Materials – HyperMiner: Topic Taxonomy Minging with Hyperbolic Embedding

**Yishi Xu, Dongsheng Wang, Bo Chen**[*]**, Ruiying Lu, Zhibin Duan**
National Laboratory of Radar Signal Processing, Xidian University, Xi'an, China
`xuyishi@stu.xidian.edu.cn, bchen@mail.xidian.edu.cn`

**Mingyuan Zhou**
McCombs School of Business, The University of Texas at Austin, USA
`mingyuan.zhou@mccombs.utexas.edu`

## A    Discussions

### A.1    Limitations

We in this paper propose a method to improve existing embedded topic models (ETMs) by introducing the hyperbolic distance to measure the semantic similarity between topics and words. Additionally, to mine a meaningful topic taxonomy with the guidance of prior structural knowledge, we further develop a regularization term based on contrastive learning that can effectively inject prior knowledge into hierarchical topic models. The main limitation of our work could be the mismatch problem between the given prior knowledge and the target corpus. Specifically, to provide proper guidance for mining an interpretable topic taxonomy, the prior structural knowledge should be well matched with the corresponding dataset. Although we present a seemingly effective heuristic strategy by finding ancestor concepts of each word in the vocabulary, there are certainly better ways to construct qualified priors to guide the learning of topic hierarchies. However, this is beyond the scope of this paper and we will conduct a thorough investigation of this issue in future work.

### A.2    Broader impact

Our work builds on advanced topic modeling techniques and thus can be used for regular text analysis. For example, topic discovery and obtaining document representation. Furthermore, our work also provides a solution to inject prior knowledge as an inductive bias to influence topic learning, which is particularly useful when users are only interested in certain types of information. Imagine a user's goal is to extract the parts about a specific topic from a large amount of news, our model can act as a good filter. Or consider the application scenario of recommending papers to researchers, the browsing history as prior knowledge reflects their preferences, which can be incorporated into the model so that only the papers on related topics are presented, thus improving the recommendation accuracy. Potential negative societal impact of our work could arise from malicious intent in changing model's behavior by injecting deliberate human prejudice, which may harm the fairness of the community. However, we hope our work is utilized to enable new downstream applications primarily from the originality of benefiting the community development.

---

[*]Corresponding author

36th Conference on Neural Information Processing Systems (NeurIPS 2022).

# B  Implementations

## B.1  Data splits

We summarize the training/test split of each dataset in Table B. 1. In particular, 20NG[2] and TMN[3] are used to evaluate both topic quality and document representation, and their document collections are divided into standard training sets and test sets. WIKI[4] and RCV2[5] are only used for topic discovery, so we use all documents for training.

Table B. 1: Splits of the datasets

|        | Vocabulary size | Number of training docs | Number of test docs |
|--------|-----------------|-------------------------|---------------------|
| 20NG   | 8,581           | 11,314                  | 7,532               |
| TMN    | 13,368          | 26,077                  | 6,520               |
| WIKI   | 20,000          | 28,472                  | /                   |
| RCV2   | 7,282           | 804,414                 | /                   |

## B.2  Prior concept taxonomy

In this part, we give an example to illustrate how the prior concept taxonomy (or structural knowledge) is constructed. Specifically, given the vocabulary of a dataset, we first filter out those words that are not included in the WordNet thesaurus. For each of the remaining words, we then find its ancestor concepts along the hypernym paths integrated in WordNet (*e.g.*, for the word *coffee*, it has a hypernym path where a series of abstract concepts, *i.e.*, *beverage*, *food*, *substance*, *physical_entity*, successively appear, as displayed in Figure B. 1). After traversing all the words, we can get a concept tree with great depth, but the number of nodes in the deepest layer may be very small. Therefore, to keep the number of nodes growing as the layer gets deeper, we choose to reserve the sevaral layers closest to the root node. For the words in the deeper layers, we connect them directly to their ancestor concepts of the deepest layer that has been preserved.

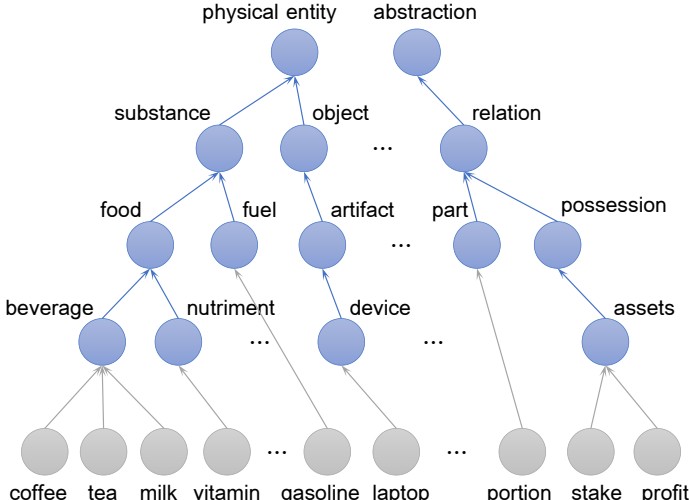

Figure B. 1: Construction of a concept taxonomy. Gray circles represent leaf nodes corresponding to the words in the vocabulary, blue circles are the ancestor nodes on the words' hypernym paths, each of which is defined by a semantic concept.

---

[2]http://qwone.com/~jason/20Newsgroups/
[3]http://acube.di.unipi.it/tmn-dataset/
[4]https://blog.salesforceairesearch.com/the-wikitext-long-term-dependency-language-modeling-dataset/
[5]https://trec.nist.gov/data/reuters/reuters.html

In this way, we construct a concept taxonomy with a depth of 5 (the layer of words is excluded) for each dataset. More precisely, the number of concepts in each layer is [2, 12, 83, 325, 560] for 20NG; [2, 11, 84, 366, 683] for TMN; [2, 12, 91, 408, 810] for WIKI; [2, 11, 70, 306, 540] for RCV2.

## B.3 Inference network

Since the exact posterior distribution for $\boldsymbol{\theta}^{(l)}$ is intractable in our generative model, we aim to design a sampling-based inference network to approximate the true posterior distribution, which is adopted by most neural topic models. In view of the hierarchical structure where deep-layer latent variables are difficult to receive effective information from the original input, we draw experience from LadderVAE [1] and use a skip-connected deterministic upward path to infer the hidden features of the input $\boldsymbol{x}$

$$
\begin{aligned}
\boldsymbol{h}_j^{(1)} &= \mathrm{MLP}\left(\boldsymbol{x}\right), \\
\boldsymbol{h}_j^{(l)} &= \boldsymbol{h}_j^{(l-1)} + \mathrm{MLP}\left(\boldsymbol{h}_j^{(l-1)}\right),
\end{aligned}
\tag{1}
$$

where MLP is a multi-layer perceptron consisting of two fully connected layers, with the ReLU activation following behind. The obtained hidden features are subsequently combined with the prior from the stochastic up-down path to approximate the variational posterior, which is expressed as

$$
\begin{aligned}
\boldsymbol{k}_j^{(l)} &= \mathrm{Softplus}\left(\mathrm{Linear}\left(\boldsymbol{\Phi}^{(l+1)}\boldsymbol{\theta}_j^{(l+1)} \oplus \boldsymbol{h}_j^{(l)}\right)\right), \\
\boldsymbol{t}_j^{(l)} &= \mathrm{Softplus}\left(\mathrm{Linear}\left(\boldsymbol{\Phi}^{(l+1)}\boldsymbol{\theta}_j^{(l+1)} \oplus \boldsymbol{h}_j^{(l)}\right)\right), \\
q\left(\boldsymbol{\theta}_j^{(l)}|\boldsymbol{h}_j^{(l)}, \boldsymbol{\theta}_j^{(l+1)}, \boldsymbol{\Phi}^{l+1}\right) &= \mathrm{Weibull}\left(\boldsymbol{k}_j^{(l)}, \boldsymbol{t}_j^{(l)}\right),
\end{aligned}
\tag{2}
$$

where $\oplus$ denotes the concatenation at topic dimension, $\mathrm{Linear}$ is a simple fully connected layer with identity activation, and $\mathrm{Softplus}$ applies $\log(1 + \exp(\cdot))$ nonlinearity to each element, ensuring that shape and scale parameters of the Weibull distribution are positive. The reason for using the Weibull distribution to approximate the gamma-distributed conditional posterior has been explained in the main body. Note that both shape and scale parameters, $i.e.$, $\boldsymbol{k}_j^{(l)} \in \mathbb{R}^{K_l}$ and $\boldsymbol{t}_j^{(l)} \in \mathbb{R}^{K_l}$, are inferred through the neural networks, by using the combination of the bottom-up likelihood information and the top-down prior information as input. Figure B. 2 depicts the overall inference process.

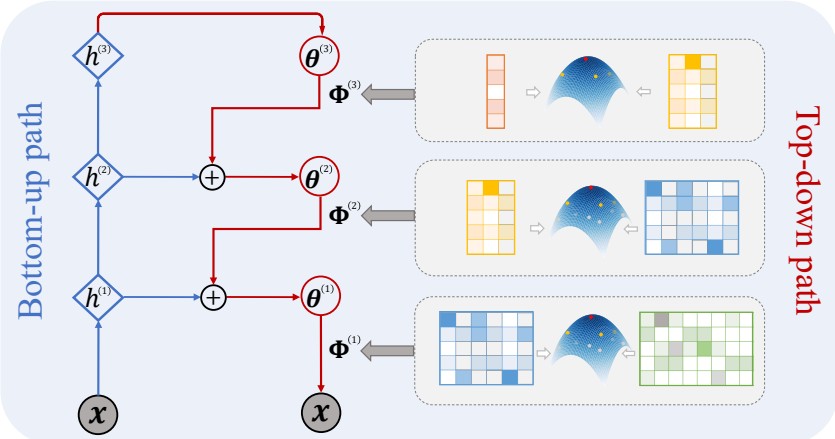

Figure B. 2: Overview of the inference network. The bottom-up path propagates the likelihood information from the original input, and the top-down path conducts the prior information from the generative model.

## B.4 Training protocal

All our experiments are performed on a single Nvidia Geforce RTX 3090 GPU card, with PyTorch as the programming platform to implement our models. For the MLP module in the inference network, we set the number of hidden neurons as 300. In addition, we also add a batch normalization layer to prevent overfitting. For all the embedded topic models, we set the embedding size as 50. To optimize our models, we use the Adam [2] optimizer with a learning rate of 0.01. As for the size of each

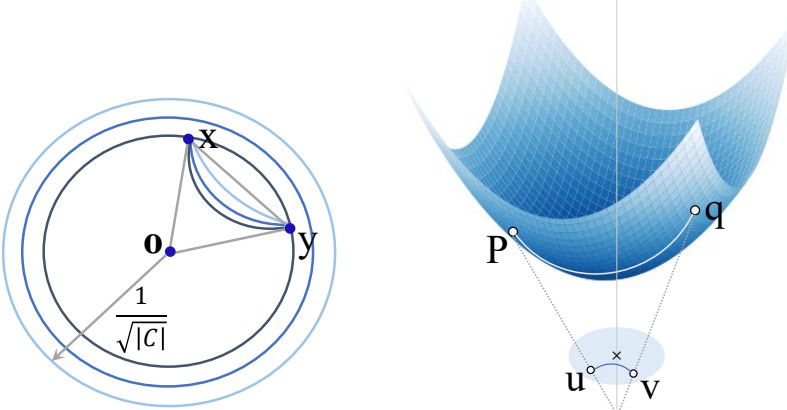

Figure C. 3: Left: Poincaré disk geodesics (shortest path) connecting x and y for different curvatures. As curvature decreases, the distance between x and y increases, and the geodesics lines get closer to the origin. Right: points (p, q) lie on the surface of a two-dimensional Lorentz space, points (u, v) are the mapping of (p, q) onto the two-dimensinal Poincaré disk. Note that (p, q) are points in three-dimensional space.

mini-batch, we set it to 200 for all datasets. What's more, for our proposed HyperMiner-KG, the size of negative samples is set as 256 for each anchor to calculate the hyperbolic contrastive loss.

It is also worth noting that the number of topics at each layer in hierarchical topic models is set to be consistent with the number of concepts at the corresponding layer in the constructed concept taxonomy. Please refer to Section B.2 for detailed settings. For the single-layer topic models, we set the number of topics to be the same as the number of concepts at the deepest layer of the concept taxonomy.

## C    Hyperbolic Space

### C.1    Equivalence between Poincaré Ball model and Lorentz model

In this section, we aim to offer an intuitive explanation about the equivalence of the two hyperbolic models mentioned in the main text. Firstly, we need to clarify the concept of geodesic. In geometry, a geodesic is commonly a curve representing the shortest path between two points in a surface. In a typical Euclidean space, the geodesic is the straight line connecting two points, and its length is the widely used Euclidean distance that is determined only by the coordinates of the two points. While in a hyperbolic space, the length of a geodesic is not only related to the coordinates of its connected points, but also affected by the curvature of hyperbolic space. This can be illustrated by the left side of Figure C. 3, as the curvature (negative) decreases, the corresponding curvature radius decreases, but the distance between x and y increases and the geodesics lines get closer to the origin.

The right side of Figure C. 3 clearly describes the projection of the geodesic on the Lorentz surface to the geodesic in the Poincaré disk. We say that the two models are mathematically equivalent because points on the Poincare disk and points in the Lorentz space can be mapped to each other, while all geometric properties including isometry are preserved. For example, to map a point in the Lorentz model into the corresponding point in the Poincaré ball, we have the following diffeomorphism [3] $p : \mathcal{H}^n \rightarrow \mathcal{P}^n$, where

$$p(x_0, x_1, \cdots, x_n) = \frac{(x_1, \cdots, x_n)}{x_0 + 1} \qquad (3)$$

Furthermore, points in $\mathcal{P}^n$ can be mapped back into $\mathcal{H}^n$ via

$$p^{-1}(x_1, \cdots, x_n) = \frac{(1 + \|x\|^2, 2x_1, \cdots, 2x_n)}{1 - \|x\|^2} \qquad (4)$$

To calculate the lengths of geodesics in the Poincaré disk and Lorentz space, respectively, please refer to the definition of $d_{\mathcal{P}}(\cdot)$ and $d_{\mathcal{L}}(\cdot)$ in Eq (**??**). However, despite the mathematical equivalence of the two models, it does not mean that the lengths calculated by $d_{\mathcal{P}}(\cdot)$ and $d_{\mathcal{L}}(\cdot)$ are exactly the same.

## C.2 Related operations

A Riemannian manifold $(\mathcal{M}, g)$ is a differentiable manifold $\mathcal{M}$ equipped with a metric tensor $g$. It can be locally approximated to a linear Euclidean space at an arbitrary point $x \in \mathcal{M}$, and the approximated space is termed as a tangent space $\mathcal{T}_x\mathcal{M}$. Hyperbolic spaces are smooth Riemannian manifolds with a constant negative curvature. There are several essential vector operations required for learning embeddings in a hyperbolic space, we will give an introduction to them in the following.

**Exponential and logarithmic maps**. An exponential map $\exp_\mathbf{x}(\mathbf{v})$ is the function projecting a tangent vector $\mathbf{v} \in \mathcal{T}_x\mathcal{M}$ onto $\mathcal{M}$. A logarithmic map projects vectors on the manifold back to the tangent space satisfying $\log_\mathbf{x}(\exp_\mathbf{x}(\mathbf{v})) = \mathbf{v}$.

**Parallel transport**. A parallel transport can move a tangent vector along the surface of a curved manifold. For example, to move a tangent vector $\mathbf{v} \in \mathcal{T}_x\mathcal{M}$ to another tangent space $\mathcal{T}_y\mathcal{M}$, we use the notation $\mathrm{PT}^\mathcal{M}_{\mathbf{x} \to \mathbf{y}}(\mathbf{v})$.

The concrete formula of these operations in Poincaré Ball and Lorentz model are summarized in Table C. 2. Where $\oplus$ and $\mathrm{gyr}[:;:]$ are the Möbius addition [4] and gyration operator [4], respectively.

Table C. 2: Summary of operations in the Poincaré ball model and the Lorentz model ($C = -1$)

| | Poincaré Ball Model | Lorentz Model |
|---|---|---|
| **Log map** | $\log^\mathcal{P}_\mathbf{x}(\mathbf{y}) = \frac{2}{\lambda_\mathbf{x}}\mathrm{artanh}\left(\lVert -\mathbf{x} \oplus \mathbf{y}\rVert\right)\frac{-\mathbf{x}\oplus\mathbf{y}}{\lVert-\mathbf{x}\oplus\mathbf{y}\rVert}$ | $\log^\mathcal{L}_\mathbf{x}(\mathbf{y}) = \frac{\mathrm{arcosh}(-\langle\mathbf{x},\mathbf{y}\rangle_\mathcal{L})}{\sqrt{\langle\mathbf{x},\mathbf{y}\rangle_\mathcal{L}^2 - 1}}(\mathbf{y} + \langle\mathbf{x},\mathbf{y}\rangle_\mathcal{L}\mathbf{x})$ |
| **Exp map** | $\exp^\mathcal{P}_\mathbf{x}(\mathbf{v}) = \mathbf{x} \oplus \left(\tanh\left(\frac{\lambda_\mathbf{x}\lVert\mathbf{v}\rVert}{2}\right)\frac{\mathbf{v}}{\lVert\mathbf{v}\rVert}\right)$ | $\exp^\mathcal{L}_\mathbf{x}(\mathbf{v}) = \cosh(\lVert\mathbf{v}\rVert_\mathcal{L})\mathbf{x} + \mathbf{v}\frac{\sinh(\lVert\mathbf{v}\rVert_\mathcal{L})}{\lVert\mathbf{v}\rVert_\mathcal{L}}$ |
| **Transport** | $\mathrm{PT}^\mathcal{P}_{\mathbf{x}\to\mathbf{y}}(\mathbf{v}) = \frac{\lambda_\mathbf{x}}{\lambda_\mathbf{y}}\mathrm{gyr}[-\mathbf{x},\mathbf{y}]\mathbf{v}$ | $\mathrm{PT}^\mathcal{L}_{\mathbf{x}\to\mathbf{y}}(\mathbf{v}) = \mathbf{v} + \frac{\langle\mathbf{y},\mathbf{v}\rangle_\mathcal{L}}{1-\langle\mathbf{x},\mathbf{y}\rangle_\mathcal{L}}(\mathbf{x} + \mathbf{y})$ |

# D Additional Results

## D.1 Effect of regularization term

To investigate the effect of the regularization term (prior structural knowledge) in HyperMiner-KG, we further evaluate the quality of document representations learned by HyperMiner-KG with different regularization coefficient $\lambda$ on document clustering tasks.

From the Table D. 3 we can see, wth the increase of the regularization coefficient, HyperMiner-KG has shown better performance on both km-Purity and km-NMI, proving that incorporating prior structural knowledge is beneficial to learning better document representations. However, the regularization coefficient is not the bigger the better, it has a most suitable value, which in our experiments is 5.

Table D. 3: km-Purity and km-NMI for document clustering, with different regularization coefficient for HyperMiner-KG. The best score of each dataset is highlighted in boldface.

| HyperMiner-KG | 20NG | | TMN | |
|---|---|---|---|---|
| | km-Purity | km-NMI | km-Purity | km-NMI |
| $\lambda = 0.1$ | $43.76 \pm 0.32$ | $42.63 \pm 0.38$ | $62.25 \pm 0.46$ | $41.32 \pm 0.49$ |
| $\lambda = 1$ | $44.13 \pm 0.33$ | $42.96 \pm 0.36$ | $62.73 \pm 0.47$ | $41.68 \pm 0.48$ |
| $\lambda = 2$ | $44.48 \pm 0.39$ | $43.28 \pm 0.41$ | $63.07 \pm 0.52$ | $42.06 \pm 0.54$ |
| $\lambda = 5$ | $\mathbf{45.16} \pm 0.35$ | $\mathbf{43.65} \pm 0.39$ | $\mathbf{63.84} \pm 0.48$ | $\mathbf{42.81} \pm 0.52$ |
| $\lambda = 10$ | $44.81 \pm 0.37$ | $43.47 \pm 0.38$ | $63.39 \pm 0.46$ | $42.34 \pm 0.50$ |