# OpenReview forum: "HyperMiner: Topic Taxonomy Mining with Hyperbolic Embedding"
_NeurIPS.cc/2022/Conference — NeurIPS 2022 Accept_

### Official Review · Reviewer_FP6y · 2022-07-10

**Rating:** 5
**Confidence:** 4
**Soundness:** 3 good
**Presentation:** 2 fair
**Contribution:** 2 fair

**Summary:**

This paper presents a neural hierarchical topic model that uses
hyperbolic embeddings. The paper uses a Gamma-Poisson factorization to
facilitate optimizing a variational objective. They augment this
objective with a taxonomy-aware contrastive loss as a way of
addressing hyperbolic manifold learning. Across four datasets, they
compare against six baselines. Averaged across multiple runs, they
demonstrate small but consistent improvements in their metrics (all
variances are fairly small).

**Questions:**

Q1. What is the impact of changing aspects of the taxonomy (more or less, or regrouping concepts)?

Q2. What is the impact of the positive and negative selection in the contrastive loss?

Q3. Please address my confusion regarding the Lorentz vs. Poincare approaches.

**Limitations:**

No: see taxonomy experiments/Q1.

**Strengths And Weaknesses:**

Update post-response: I thank the authors for their answers to my questions.

I understand there's a limited amount that can be done during the author response. That said, while the additional taxonomy experiments are encouraging, though for inclusion in a paper they would need additional discussion.

For whatever revision is next, I suggest that while the proposed clarifications of Poincare vs. Lorentz may be helpful for some readers, the main question is, "is this level of detail necessary for the main paper?" If the answer is yes, then this point needs to be made more immediately obvious.


=========================

Overall, this paper has a clear mathematical overview. For the
singular main contribution (hyperbolic hierarchical topic modeling)
this paper presents a decent amount of experimental evidence that the
proposed approach is effective on purely quantitative measures. While
this paper does not address the impact of the proposed approach on the
topic model as a language model (perplexity), the classification
results give some evidence as to the effect on document modeling.

However, lack of examples and experimentation when it comes to the
taxonomy is noticable. This is especially the case given the tied
nature of the model to the taxonomy (L167-169). This is important not
just for overall completeness, but to better understand the results
that have been presented. The gains, e.g., in Figure 3, are generally
small but consistent. That alone is not a negative. However, those
smaller gains are less grounded without additional examples or
experimentation.

This paper has some reorganization and other presentation
issues. While these issues could arguably be viewed as "easily
addressable," they do impact the ability of a reader to udnerstand
what the paper's take-aways are.

* The current organization of the paper suggests that the authors
  apply hyperbolic embeddings to a Poisson Gamma belief network
  only. However, as the results show, this is not the case. I
  recommend reorganizing so that the main aspect of "hyperbolic
  embeddings applied to (hierarchical) topic models" stands out.

* Since Figure 3 only makes distinctions based on color, it is a bit
  difficult to quickly interpret.

* The mathematical introduction of both the Lorentz and Poincare
  similarities is confusing and, given the amount of space dedicated
  to the equivalence explanation (L140), too nuanced. Indeed, even
  from the appendix this equivalence is not clear (am I missing
  something obvious)?

* While Fig. 4b is helpful, 4a is too busy. The concept hierarchy is
  clear (but that's a given), while the lexical hierarchy, except at a
  very coarse granularity, is not.

---

> ### Author Response · Authors · 2022-08-02
> **Detailed response to Reviewer FP6y's questions (1/2)**
>
> We sincerely appreciate your valuable comments and suggestions. As you mentioned that the gains in Figure 3 are generally small, we will add corresponding examples to visualize the content of learned topics. In addition, we will try our best to address the reorganization and presentation issues in the revision based on your advice. And your main questions have been addressed as below.
>
> **Q1:**
> What is the impact of changing aspects of the taxonomy (more or less, or regrouping concepts)?
>
> **A1:**
> To investigate the impact of changing aspects of the taxonomy, we conducted complementary experiments. Specifically, in the manuscript we use a concept taxonomy with a depth of 5  for all experiments. Here we additionally consider three concept taxonomies with depths setting as 3, 4, and 6, respectively. The following tables show the document clustering performance and topic quality of HyperMiner-KG guided by taxonomies of different scales.
>
> **Table 1:** performance on topic coherence. (Dataset: 20NG)
> | **Taxonomy Structure** | **Layer 1** | **Layer 2** | **Layer 3** | **Layer 4** | **Layer 5** | **Layer 6** |
> |------------------------|-------------|-------------|-------------|-------------|-------------|-------------|
> | 83-12-2                | 0.4180   | 0.2893   | 0.2043   | -             | -              | -             |
> | 325-83-12-2            | 0.4004   | 0.3788   | 0.2667   | 0.1878   | -              | -             |
> | 560-325-83-12-2        | 0.3160   | 0.3568   | 0.3223   | 0.2561   | 0.1925   | -             |
> | 620-560-325-83-12-2    | 0.2329   | 0.2440   | 0.2445   | 0.2614   | 0.2154   | 0.1954   |
>
> **Table 2:** performance on topic diversity. (Dataset: 20NG)
> | **Taxonomy Structure** | **Layer 1** | **Layer 2** | **Layer 3** | **Layer 4** | **Layer 5** | **Layer 6** |
> |------------------------|-------------|-------------|-------------|-------------|-------------|-------------|
> | 83-12-2                | 0.8200   | 0.9400   | 1.0000   | -           | -           | -           |
> | 325-83-12-2            | 0.5921   | 0.8124   | 0.9267   | 0.9600   | -           | -           |
> | 560-325-83-12-2        | 0.2278   | 0.3896   | 0.5833   | 0.7162   | 0.8000   | -           |
> | 620-560-325-83-12-2    | 0.1681   | 0.0890   | 0.1245   | 0.1540   | 0.3568   | 0.7400   |
>
> **Table 3:** performance on two evaluation metrics for the clustering task. (Dataset: 20NG)
> | **Taxonomy Depth** | 3 | 4 | 5 | 6 |
> |--------------------|---|---|---|---|
> | **km-Purity**      | 0.4325  | 0.4637  | 0.4498  | 0.4269  |
> | **km-NMI**         | 0.4152  | 0.4473  | 0.4331  | 0.4286  |

---

> > ### Author Response · Authors · 2022-08-02
> > **Detailed response to Reviewer FP6y's questions (2/2)**
> >
> > **Q2:**
> > What is the impact of the positive and negative selection in the contrastive loss?
> >
> > **A2:**
> > In the standard contrastive loss, each example picks the positive sample from one of its own augmented views and selects other examples in the mini-batch as negative samples. Generally, the number of negative samples can have a noticeable impact on performance. Therefore, in our hyperbolic contrastive loss, we keep using each node's first-order neighbors as its positive samples, but select a different number of negative samples from the non-first-order neighbors to study its impact.
> > The comparative results are presented below.
> >
> > **Table 4:** performance on document clustering. (Dataset: 20NG)
> > | **Number of negative samples** | **km-Purity** | **km-NMI** |
> > |--------------------------------|---------------|------------|
> > | 128                            | 0.4237         | 0.4240   |
> > | 256                            | 0.4498         | 0.4331   |
> > | 512                            | 0.4323         | 0.4285   |
> > | All non-first-order neighbors | 0.4522   | 0.4496  |
> >
> > **Q3:**
> > Please address my confusion regarding the Lorentz vs. Poincare approaches.
> >
> > **A3:**
> > We acknowledge that the equivalence between Poincaré and Lorentz similarities may not be well understood from the mathematical formulation alone, so we refer to Figure 1 of "*Nickel, M. and Kiela, D., Learning continuous hierarchies in the Lorentz model of hyperbolic geometry, ICML 2018*" for an intuitive understanding. Figure 1b clearly describes the projection of the geodesic on the Lorentz surface to the geodesic in the Poincaré disc, and the length of the geodesic (the shortest distance between two points) in the Poincaré disc is defined by $d_{\mathcal{P}}(x, y)$ in Equation (1), the length of the geodesic on the surface of a Lorentz space is given as $d_{\mathcal{L}}(x, y)$ in Equation (5). These formulations are followed by our manuscript. Since the points in the Poincaré disc and the points in the Lorentz space can be mapped to each other (all geometric properties including isometry are preserved), we say the two models are mathematically equivalent. However, it does not mean that the actual lengths calculated by $d_{\mathcal{P}}(x, y)$ and $d_{\mathcal{L}}(x, y)$ are exactly the same. We will also add the corresponding diagram in the revision to help readers understand this better.

---

### Official Review · Reviewer_VQec · 2022-07-11

**Rating:** 7
**Confidence:** 4
**Soundness:** 4 excellent
**Presentation:** 4 excellent
**Contribution:** 3 good

**Summary:**

This paper improves embedded topic modeling by using a hyperbolic space. The proposed approach particularly helps considering hierarchical structure of words and topics. In addition, the authors propose a revised contrastive loss to inject prior knowledge. The evaluation compares the proposed approach with 6 existing embedding and non-neural-embedding-based topic modeling approaches on 4 datasets. The proposed approach is shown to outperform the existing approaches in most configurations. Some analyses including the 2D visualization show the proposed approach performs as intended and encode the hierarchical information.

**Questions:**

The paper is quite straightforward and I appreciate it. Please take a look at W1-W3 for suggestions to further improve the presentation.

**Limitations:**

I cannot spot any negative societal impact of this work.

**Strengths And Weaknesses:**

Strengths
- S1. The proposed approach encodes hierarchical information, and can incorporate the existing knowledge by adopting the hyperbolic space into an existing neural topic modeling approach.
- S2. The paper has good presentation and readability in general.
- S3. The solid empirical results show the performance of the constructed topic model and the analysis shows that the approach works as intended.

Weaknesses
- W1. The variants HyperETM, HyperMiner are HyperMiner-KG are not explained, and it thus reduces the reproducibility of the experiments.
- W2. The topic taxonomy (Figure 4b) is hard to relate without document examples and/or summary of the topics.
- W3. Discussion of side-effect is missing. As the hyperbolic space is enforced, the freedom of representing a topic decreases. As such, there can be side-effects and a discussion of desired properties in topic modeling and side-effects from this approach would be beneficial. For example, is the tree-structure the best or can it be a rather limiting factor?

---

> ### Author Response · Authors · 2022-08-02
> **Detailed response to Reviewer VQec's concern on the presentation**
>
> We appreciate your constructive comments and suggestions, which are helpful for us to further improve the presentation quality of our paper. The weaknesses have been addressed as follows.
>
> **W1:**
> The variants HyperETM, HyperMiner and HyperMiner-KG are not explained, and it thus reduces the reproducibility of the experiments.
>
> **A1:**
> We apologize for not explaining the variants HyperETM, HyperMiner and HyperMiner-KG in the manuscript. HyperETM and HyperMiner are variants of ETM and SawETM, respectively. In both variants, words and topics are embedded into a shared hyperbolic space, so that the semantic hierarchy among words and topics can be better captured by the distance between their embeddings. Besides,
> HyperMiner-KG is developed based on HyperMiner to inject external prior structural knowledge through our proposed hyperbolic contrastive loss. We will add these explanations in the revision.
>
> **W2:**
> The topic taxonomy (Figure 4b) is hard to relate without document examples and/or summary of the topics.
>
> **A2:**
> We agree with you that Figure 4b could be difficult to interpret without a summary of the topics. Since each topic in our model (HyperMiner-KG) is learned under the guidance of a prior concept, we can annotate the name of the corresponding concept to evaluate whether the learned topic is successfully influenced by the prior concept. This issue will be addressed in our revision.
>
> **W3:**
> Discussion of side-effect is missing. As the hyperbolic space is enforced, the freedom of representing a topic decreases. As such, there can be side-effects and a discussion of desired properties in topic modeling and side-effects from this approach would be beneficial. For example, is the tree-structure the best or can it be a rather limiting factor?
>
> **A3:**
> The side-effect discussion is meaningful and necessary. Indeed, the freedom of representing a topic might somewhat decrease as our approach enforces the hyperbolic space, but we believe that the capacity of such a space is sufficient for topics to learn good representations. More importantly, the freedom of representing every single topic may not be a decisive factor for the performance in topic modeling tasks. Instead, the relative distance between topics and words can be more significant since we expect it to well reflect the semantic relatedness, which is critical for learning interpretable topics. In view of the inherent semantic hierarchy of words and the topic hierarchy in hierarchical topic modeling, a distance function with the property of expressing such hierarchical relationships would be better. And the distance metric in hyperbolic space just meets this need, our experimental results also demonstrate that hyperbolic space is suitable for the task of topic modeling. Further, considering that semantic relations include not only hypernymy but also antonymy and meronymy, the tree-structure may not be the best, but at least it is effective in hierarchical topic modeling.

---

> > ### Comment · Reviewer_VQec · 2022-08-09
> > **Thanks for the response**
> >
> > I appreciate the authors' answers to my questions, and I agree with those, and hope that can improve the paper a bit more. The discussion on the limitation is on point, and I thought the same about antonymy and meronymy, so a caveat would be this might need to be used together with another embedding if those are important for a certain domain. Clarifying this limitation would be helpful to future readers, but this paper does a solid job within the proposed problem.

---

### Official Review · Reviewer_BaAH · 2022-07-12

**Rating:** 7
**Confidence:** 3
**Soundness:** 4 excellent
**Presentation:** 4 excellent
**Contribution:** 4 excellent

**Summary:**

This paper studies the hierarchical topic modeling problem. Existing studies generally learn topic representations in the Euclidean space, which might lead to some limitations in capturing hierarchical relations. To be more specific, the ability to model complex patterns is inherently bounded by the dimensionality of the embedding space. On the other hand, side information such as taxonomy of concepts is sometimes available to guide the learning of hierarchical topics, which might be challenging to preserve in the Euclidean space. As a consequence, this paper proposes a novel framework that introduces learning word and topic representations in the hyperbolic space. Experiments on three public text datasets and auxiliary ablation/case studies demonstrate the effectiveness of the proposed framework.

**Questions:**

1. If the ability of Euclidean space is bounded by its dimensionality, then what interests the reviewer is that if the dimensionality of one hyperbolic space is d, then what the dimension would be for a Euclidean space to achieve comparable performance in the topic modeling task (e.g., $k * d$ where $k$ is a fixed number)?

**Limitations:**

As mentioned in the appendix, the main limitation of this work is the mismatch problem between the given structural knowledge and the target corpus. To provide proper guidance for mining an interpretable topic taxonomy, the prior structural knowledge should be well matched with the corresponding dataset.

**Strengths And Weaknesses:**

Strengths:
1. This paper studies an important task. Hierarchical topic modeling is a well-studied yet important problem that has the potential of benefiting a wide spectrum of downstream applications such as classification, named entity recognition, etc.
2. The motivation of the solution is very clear. As is well known, Euclidean space falls short when it comes to modeling hierarchical structures. In contrast, hyperbolic space has nice properties of hierarchical awareness and spacious room which benefits hierarchical relationship modeling.
3. The experiments are thorough. They are sufficient to justify the superiority of hyperbolic space under the problem set. An ablation study between HyperMiner and HyperMiner-KG validates the modeling of hierarchical external knowledge. Meanwhile,  the experiments also show that the proposed method is extendible to all kinds of ETM models.
4. The paper is carefully written and well organized.

Weaknesses:
1. More related works could be potentially included and discussed [1].

[1] Hierarchical Topic Mining via Joint Spherical Tree and Text Embedding

Minor issues:
- Typos
In line 137, "Sine" should be "Since"
In Appendix line 93, "wth" should be "with"

---

> ### Author Response · Authors · 2022-08-02
> **Detailed response to Reviewer BaAH's question**
>
> We appreciate your detailed comments and suggestions. We will take your advice to include more discussions of related work and fix all identified typos in the manuscript. Regarding the question of interest to you, we make the following statements.
>
> **Q1:**
> If the ability of Euclidean space is bounded by its dimensionality, then what interests the reviewer is that if the dimensionality of one hyperbolic space is $d$, then what the dimension would be for a Euclidean space to achieve comparable performance in the topic modeling task (e.g., $k \times d$ where $k$ is a fixed number)?
>
> **A1:**
> We acknowledge that it can be difficult to theoretically analyze the dimension required for Euclidean space to achieve comparable performance when the dimensionality of hyperbolic space is $d$. Although we can mathematically show that hyperbolic space has the advantage of a more spacious room than Euclidean space (e.g., the area of a circle in two-dimensional Euclidean space is $2\pi r^2$ and the disc area in a two-dimensional hyperbolic space is given as $\pi(e^{-r} + e^{r})$, indicating that the area expands exponentially with $r$ in hyperbolic space while grows polynomially in Euclidean space), this does not guarantee better performance on topic modeling tasks. Empirically, however, we may gain some insights from comparative experiments. In our manuscript, Table 3 presents the comparison results of the two spaces in the document classification task under different embedding dimension settings. Here we give additional results on the quality of learned topics in the two spaces of different dimensionalities.
>
> **Table 1:** comparative performance on topic coherence. (Dataset: 20NG)
> | **Method**             | **Dimensionality** | **Layer 1** | **Layer 2** | **Layer 3** | **Layer 4** | **Layer 5** |
> |------------------------|--------------------|-------------|-------------|-------------|-------------|-------------|
> |                    | 2                  | 0.1770      | 0.1251      | 0.1031      | 0.0470      | 0.0563      |
> | **SawETM**             | 5                  | 0.1800      | 0.2379      | 0.1547      | 0.1095      | 0.0495      |
> | **(Euclidean space)**  | 20                 | 0.2265      | 0.2488      | 0.1744      | 0.1398      | 0.0654      |
> |                    | 50                 | 0.3044      | 0.2729      | 0.1843      | 0.1634      | 0.0679      |
> |                    | 2                  | 0.1800       | 0.1729        | 0.1388        | 0.1587      | 0.1871      |
> | **HyperMiner**         | 5                  | 0.2004      | 0.2654      | 0.2618      | 0.2005      | 0.1763      |
> | **(Hyperbolic space)** | 20                 | 0.2440     | 0.3072     | 0.2659      | 0.2472      | 0.1836      |
> |                    | 50                 | 0.2954      | 0.3437      | 0.3101      | 0.2753      | 0.1843      |
>
> **Table 2:** comparative performance on topic diversity. (Dataset: 20NG)
> | **Method**             | **Dimensionality** | **Layer 1** | **Layer 2** | **Layer 3** | **Layer 4** | **Layer 5** |
> |------------------------|--------------------|-------------|-------------|-------------|-------------|-------------|
> |                    | 2                  | 0.0085      | 0.0107      | 0.0188      | 0.1200      | 0.5800      |
> | **SawETM**             | 5                  | 0.0579      | 0.0607      | 0.0857      | 0.1600      | 0.5200      |
> | **(Euclidean space)**  | 20                 | 0.1551      | 0.1945      | 0.2632      | 0.3546      | 0.6800      |
> |                    | 50                 | 0.1706      | 0.2367      | 0.3809      | 0.5471      | 0.7400      |
> |                    | 2                  | 0.0137        | 0.0166        | 0.0241        | 0.1500      | 0.6200      |
> | **HyperMiner**         | 5                  | 0.0596      | 0.0831      | 0.1838      | 0.3400      | 0.5600      |
> | **(Hyperbolic space)** | 20                 | 0.1684      | 0.2148      | 0.2971      | 0.4733      | 0.7600      |
> |                    | 50                 | 0.1987      | 0.3679      | 0.5133      | 0.6362      | 0.7800      |

---

### Meta-Review · Area_Chair_EZDP · 2022-08-29

**Recommendation:** Accept
**Confidence:** Certain

**Metareview:**

Hyperbolic embeddings were a fascinating alternative to Euclidean embeddings that never seemed to take off, despite having significant conceptual advantages in representing the oddities of semantics. I am happy to see more work on curved spaces as a tool for semantic analysis! This work has strong reviews, and reviewers were generally happy with the author responses. I'd like to see it published.

**Award:**

No

---

### Decision · Program_Chairs · 2022-09-14

Accept